# DETECTING LABEL NOISE FROM LOSS CURVES

## ABSTRACT

This paper introduces a new method to discover mislabeled training samples and to mitigate their impact on the training process of deep networks. At the heart of our algorithm lies the Area Under the Loss (AUL) statistic, which can be easily computed for each sample in the training set. We show that the AUL reliably reveals differences in training dynamics between (clean) samples that benefit from generalization and (mislabeled) samples that need to be "memorized". We demonstrate that the estimated AUL score conditioned on clean vs. noisy is approximately Gaussian distributed and can be well estimated with a simple Gaussian Mixture Model (GMM). The resulting GMM provides us with mixing coefficients that reveal the percentage of mislabeled samples in a data set as well as probability estimates that each individual training sample is mislabeled. We show that these probability estimates can be used to down-weight suspicious training samples and successfully alleviate the damaging impact of label noise. We demonstrate on the CIFAR10/100 datasets that our proposed approach is significantly more accurate and consistent across model architectures than all prior work.

## 1  INTRODUCTION

The success of supervised machine learning is inherently dependent on the reliability of the labels in the training data. Most algorithms assume that the training labels are clean and can be trusted blindly. However, creating real-world data sets that do not suffer from substantial label noise is extremely challenging. Human annotators, especially on crowdsourced platforms, can be prone to making labeling mistakes. "Weakly-labeled" datasets (Xiao et al., 2015; Joulin et al., 2016; Li et al., 2017a), which collect labels through proxy variables or web scraping, are especially likely to mislabeled samples. In fact, even the celebrated MNIST (LeCun & Cortes, 2005) and ImageNet (Deng et al., 2009) data sets famously contain multiple mislabeled images (see Figure 1 for suspicious examples detected by our proposed method). In practice, it is often too costly to perform manual curation or quality control to remove such examples on large datasets.

In this paper we develop a framework that provides multiple ways to handle label noise. Our method exploits the recent paradigm of training deep neural networks via stochastic gradient descent (SGD) (Huang et al., 2019; He et al., 2016). These overparameterized networks can achieve zero training error even when labels are assigned randomly (Zhang et al., 2017) — essentially *memorizing* the entire data set. Contrary to classical machine learning wisdom, subsequent empirical and theoretical analyses (Arpit et al., 2017; Allen-Zhu et al., 2019; Arora et al., 2019) show that despite their capacity to overfit, these models can achieve impressive generalization performance. The implicit regularization of SGD permits memorization only if it is necessary to classify a particular training point correctly. We refer to this (latent) variable as the *memorization need* of a sample.

Although prior works often attempt to detect label noise through various methods involving validation loss on rotating hold-out sets (Chen et al., 2019), training auxiliary neural networks (Xiao et al., 2015; Jiang et al., 2017), or meta-learning (Ren et al., 2018), we argue that the model's training dynamics can be far more revealing. For example, data points with high memorization need will exhibit noticeably different loss trajectories than those with low memorization need (see Figure 2).

We capture these differences in loss trajectories by introducing a simple, informative training statistic: the *Area Under the Loss Curve* (AUL). As its name suggests, the AUL can be trivially computed for a whole training set or any single training sample. It smooths the loss trajectory across all training epochs, capturing the general learning behavior of the samples for which it is computed. We

MNIST: High AUL Images          ImageNet-100 High AUL Images

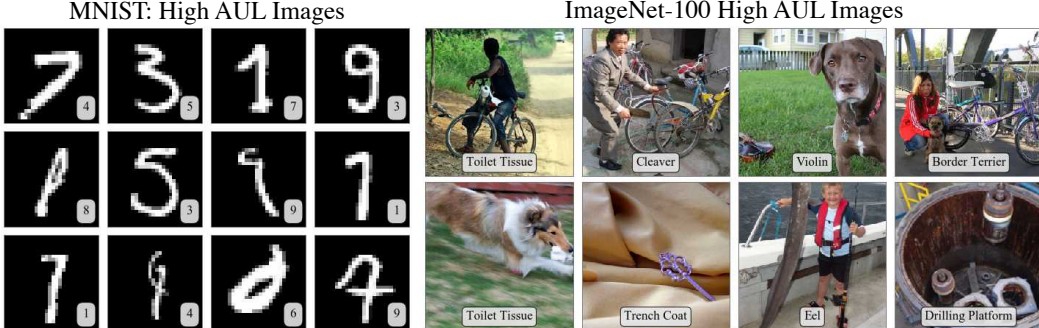

Figure 1: Images from MNIST (left) and the first 100 classes of ImageNet (right) with the largest *Area Under the Loss Curve* (AUL). The high-AUL images are mislabeled or extremely ambiguous. MNIST AULs are from a LeNet model and ImageNet AULs are from a DenseNet-121 model.

observe that in the presence of label noise, the distribution of AUL scores tends to be bimodal and can be estimated with a 1-dimensional, 2-component Gaussian mixture model (GMM). This provides us with three useful ways to manage label noise: a) the mixing coefficients naturally estimate the fraction of noisy samples in the training data set; b) the membership likelihood of the higher-mean Gaussian estimates the probability that a sample is mislabeled; c) the clean-label probabilities can be used to weight samples in order to mitigate the influence of label noise in training.

Despite its compelling simplicity (the AUL score and GMM fit require only a few lines of Python), it is currently by far the most reliable and accurate estimator of label noise. We evaluate our method on standard benchmark datasets with 2 models of synthetic label uncertainty. We demonstrate empirically that our method well-approximates the overall percentage of noisy data and detects mislabeled points with high recall across a wide range of noise levels. Furthermore, we use AUL to achieve state-of-the-art results of 11.1% and 40.2% classification error with 40% class-dependent label noise on CIFAR10 and CIFAR100, respectively. Finally, we show that the AUL statistic is consistent across network architectures and training runs.

## 2 RELATED WORK

**Learning with Weak or Noisy Labels.** One popular approach to mitigate the effects of label noise is to design more robust training objectives. Zhang & Sabuncu (2018) propose a novel loss function by generalizing the cross entropy loss. Others interpolate target labels with predicted labels (Reed et al., 2014; Ma et al., 2018) or modify the loss term with an estimated noise transition matrix (Sukhbaatar et al., 2015; Patrini et al., 2016; Hendrycks et al., 2018). Li et al. (2017b) train a network on a small set of trusted data and guide learning through knowledge distillation. Another line of work either reweights or filters training samples to discourage learning from mislabeled examples. Attempted metrics to select training data includes training loss (Han et al., 2018; Shen & Sanghavi, 2019), test/validation loss (Chen et al., 2019), and gradient information on clean samples (Ren et al., 2018). MentorNet (Jiang et al., 2017) learns a meta-network to output example weights. Malach & Shalev-Shwartz (2017) train two networks and identify examples with prediction discrepancies. Our work also mitigates the effects of mislabeled samples through sample weights; however, in contrast to most prior work we extract these weights from the training dynamics.

**Analysis of Neural Network Training.** Recently, Neyshabur et al. (2019) and Allen-Zhu et al. (2019) both bound the generalization error of overparameterized neural networks assuming uncorrupted labels. Another line of work analyzes the training dynamics of SGD under various assumptions on the loss and network architecture (Mandt et al., 2017; Zou et al., 2019; Li & Liang, 2018). Several recent works attempt to explain the observation that deep networks can overfit to random training labels (Zhang et al., 2017). For example, Arpit et al. (2017) measure the gradient norms across epochs and empirically show that the number of clean data points with high loss sensitivity decreases throughout training. Arora et al. (2019) prove generalization bounds and convergence rates of GD for two-layer ReLU networks as a function of the training data and labels. Other recent papers prove that GD combined with either early stopping (Li et al., 2019) or regularization (Hu et al., 2019) generalizes robustly in the presence of label noise. These analyses provide both the-

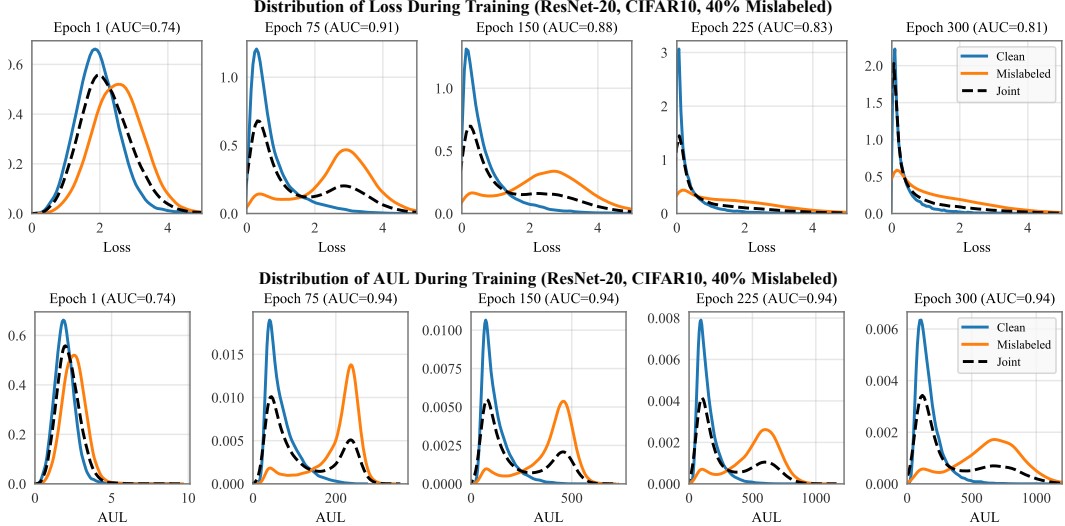

Figure 2: A comparison of training loss and training AUL as proxies for identifying mislabeled samples. **Top:** Histograms of training sample loss over the course of training. The distributions for clean and mislabeled samples are very separable at the beginning of training, but become less separable towards the end of training. **Bottom:** Histograms of training sample AUL. The distributions for clean and mislabeled samples remain separable even after the network has converged.

oretical and empirical evidence that deep networks follow a two-stage training procedure: models first learn generalizable parameters on the clean data before memorizing the noisy samples.

## 3 THE LOSS CURVE AND MEMORIZATION NEED

In this section we demonstrate that training loss dynamics can be used to identify data points with high memorization need. We generate classification datasets with various amounts of memorization need by changing the labels of a subset of data points. We then introduce a metric – Area Under the Loss Curve (AUL) – to identify these mislabeled (*i.e.* high memorization need) points. The AUL metric exhibits several desirable properties: high detection accuracy, stability over the course of training, and invariance to model architecture.

**Rate of loss reduction indicates memorization need.** To illustrate the training dynamics of high memorization need data, we train a ResNet-20 on the CIFAR10 dataset where $40\%$ of the training labels have been replaced with random labels. Following standard practice Huang et al. (2017), we use data augmentation, which is why the training loss does not diminish to zero for all samples. The learning rate is initialized to $0.1$ and reduced by a factor of 10 at epochs 150 and 225. Since there is no relation between the incorrect labels and their corresponding images, these data points by design have a high memorization need. In Figure 2 (top) we plot histograms of the training loss at various stages of training for both the correctly and incorrectly labeled data points. We make several observations. First, after the network has converged (epoch 300, top right histogram), most data points (both correctly and incorrectly labeled) have low training loss. This result is consistent with the observations of Zhang et al. (2017), suggesting that the network is memorizing these incorrectly labeled points. More notably however are the loss curves at earlier stages of training. At 75 and 150 epochs (top middle-left and top middle histograms) the training loss for mislabeled points is much higher than that of the correctly labeled points. In fact, the incorrectly labeled points tend to have a larger loss than at the beginning of training (epoch 1, top left histogram). Consistent with Arpit et al. (2017); Arora et al. (2019); Li et al. (2019), this suggests that although all points achieve low loss, *samples with high memorization need require more training iterations to achieve low loss.*

**Capturing rate of loss change through AUL.** Given that loss dynamics are indicative of the memorization need, our goal is to capture this signal with a robust metric. One challenge is that loss curves for individual samples are rather noisy as a result of stochastic optimization. In Figure 3 (left)

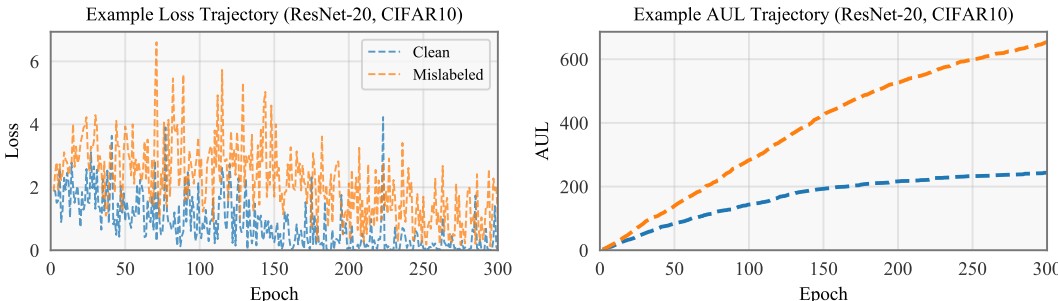

Figure 3: **Left:** Loss trajectories over epochs for a single clean sample and a single mislabeled example (randomly selected). On average, the clean sample has a lower loss than the mislabeled example. However, the loss decreases over time for both examples, and the trajectories are noisy and non-monotonic. **Right:** AUL trajectory for the same examples. Integrating loss over training time reduces trajectory noise, making the samples' different learning dynamics more separable.

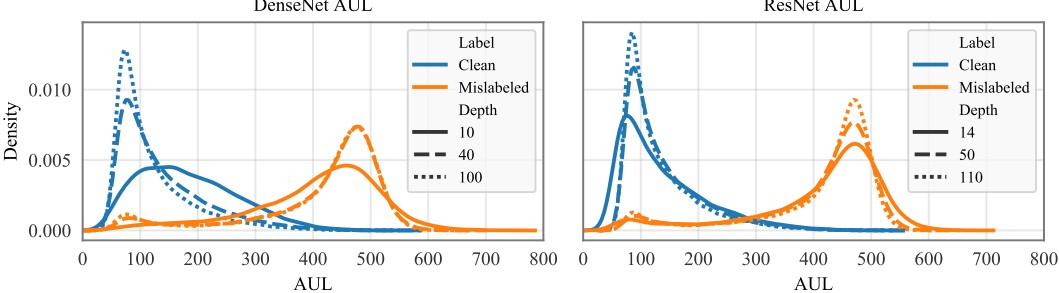

Figure 4: Distribution of AUL (CIFAR10, 40% label noise) computed on ResNets and DenseNets of different depth. The modes of the distribution are consistent across networks.

we plot loss curves for two example training points – one clean and one mislabeled. The variance of the loss is quite high, and there are several training epochs where the mislabeled point achieves lower loss than the correctly labeled point. The second challenge is that training loss becomes less indicative of memorization need in later epochs. In Figure 2 we observe that all samples have low training loss by epoch 300. To assess the discriminating ability of the loss, we compute the Area under the ROC Curve (AUC) at various training stages. Although the final training loss still separates clean and mislabeled samples with an AUC of $81\%$, it is far less separable than at earlier epochs. In other words, the signal in the training loss varies throughout and becomes weaker over time.

One simple way to improve the signal to noise ratio is to integrate training loss over time. As seen in Figure 3, the losses of correctly-labeled samples decay early during training while the mislabeled samples' losses do not decay until later epochs. Therefore, clean samples will *in expectation* have a lower loss than mislabeled samples at any given epoch. We formalize this intuition in a metric which we refer to as *Area Under the Loss Curve*, or AUL. Formally, for a given sample $(\mathbf{x}, y)$,

$$\mathrm{AUL}_{\mathbf{x},y} = \sum_{i=1}^{T} \mathcal{L}\left(\phi_i(\mathbf{x}), y\right),$$

where $T$ is the total number of epochs, $\mathcal{L}\left(\cdot, \cdot\right)$ is the loss function, and $\phi_i$ is the model at epoch $i$. In Figure 3 (center, right) we observe that the AUL dynamics are far less noisy than loss dynamics. The clean sample achieves a lower AUL than the mislabeld sample at every epoch. Moreover, the curves remain separable even after the loss signal decays in later epochs. Figure 2 (bottom row) shows the AUL histograms of clean and mislabeled samples at several stages of training. The AUC score of the AUL is $94\%$ at epoch 75 and remains this high through epoch 300. This contrasts with the loss, which reaches its best predictive accuracy at epoch 75 (AUC=0.91) and decreases thereafter. While AUL remains a separable metric throughout the whole training cycle (including learning rate drops), in practice we find it sufficient to compute the AUL before the first scheduled learning rate drop.

**Stability and consistency of AUL and training dynamics.** Unlike test-time predictions, which vary with network architecture and even with network initialization, training dynamics are relatively

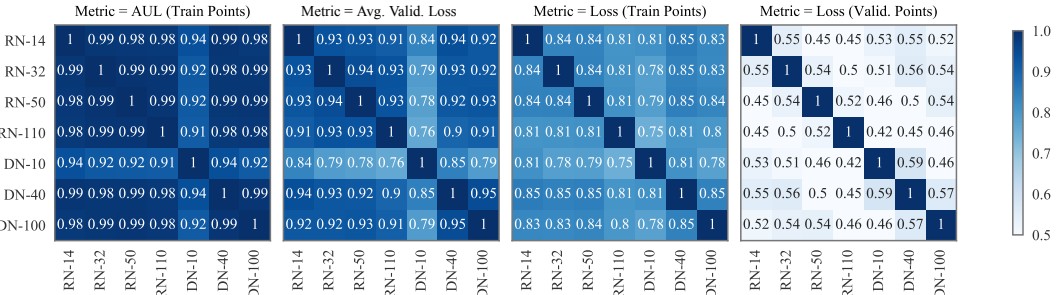

Figure 5: Pearson's correlation of AUL and other metrics across various network architectures (RN=ResNet, DN=DenseNet) after 150 epochs. AUL produces a very consistent ranking of the training data. Training loss produces less consistent rankings. Furthermore, the loss on held-out data points (from the test set) is also much less correlated across networks than AUL.

consistent across networks. In Figure 4 we plot the AUL distribution after training ResNet and DenseNet models of various depth (up to the first learning rate drop). Each network is trained on the CIFAR10 dataset with standard data augmentation, where the same 40% of samples are mislabeled. The AUL distributions are remarkably similar for networks of different depths. The modes are more sharply peaked for deeper networks, yet the modes are around roughly the same AUL values. To further demonstrate the consistency of training dynamics, we compute the Pearson's correlation coefficient of the dataset's AUL scores computed between different network architectures (Figure 5, left). Most pairs of networks achieve a correlation of 0.98 or 0.99. These results suggest that *the AUL metric captures properties that are dataset-dependent rather than model-dependent.*

Conversely, the testing predictions of different models are less correlated than training dynamics. In Figure 5 (middle left) we measure the inter-network correlation of loss validation set samples. We average each sample's loss over all training epochs which – similarly to AUL – reduces the impact of noise from any one epoch. The averaged validation loss achieves correlations between 0.8 and 0.95. While this is still a high correlation, we note that it is not as consistent as AUL. Finally, we observe that integrating loss values over time improves the consistency of the metric. Figure 5 (middle right) plots the inter-network correlation training loss at epoch 150 (the last epoch before the learning rate is dropped). Most networks achieve a correlation between 0.79 and 0.85 – roughly 10 percentage points less than training AUL. There is even less inter-network correlation for validation loss (Figure 5, right). At epoch 150 the correlation is on average 0.55, which is roughly 40 percentage points less than averaged validation loss. The loss of an individual sample at any given epoch is likely impacted by stochastic training noise, and integrating reduces this noise.

## 4 INCORPORATING AUL DYNAMICS INTO ROBUST TRAINING

Though AUL is relatively invariant to network architecture choices, its range depends on both the dataset and the training length. Next we introduce simple algorithms that convert the relative AUL metric into interpretable probabilities for (a) estimating the amount of noise in a dataset, (b) identifying which samples are likely to be mislabeled, and (c) reweighting samples to reduce overfitting.

### 4.1 ESTIMATING NOISE AND IDENTIFYING MISLABELED SAMPLES

For $\mathbf{x}$, a given input from the training dataset, let $y$ be its ground-truth label (which is unobserved), and let $\widetilde{y}$ be the potentially incorrect observed label. We assume that the observed labels in the dataset follow some unknown noise model $p(\widetilde{y} \mid y)$. One goal is to estimate the amount of label noise in the training dataset: *i.e.* $p(\widetilde{y} \mid y)$. A more challenging goal is identifying which samples are most likely to be mislabeled. More specifically, we want to estimate $p(\widetilde{y} \neq y \mid \mathbf{x})$ for all $\mathbf{x}$.

As mentioned in the previous section, AUL is a relative metric and so its values cannot be directly converted into a probability $p(\widetilde{y} \neq y \mid \mathbf{x})$ that a sample is mislabeled. Therefore, we instead estimate the conditional AUL distributions for clean and mislabeled samples, *i.e.* $p(\mathrm{AUL}_{\mathbf{x},\widetilde{y}} \mid y = \widetilde{y})$ and $p(\mathrm{AUL}_{\mathbf{x},\widetilde{y}} \mid y \neq \widetilde{y})$ respectively. $p(y \neq \widetilde{y} \mid \mathrm{AUL}_{\mathbf{x},\widetilde{y}})$ is estimated from these conditional

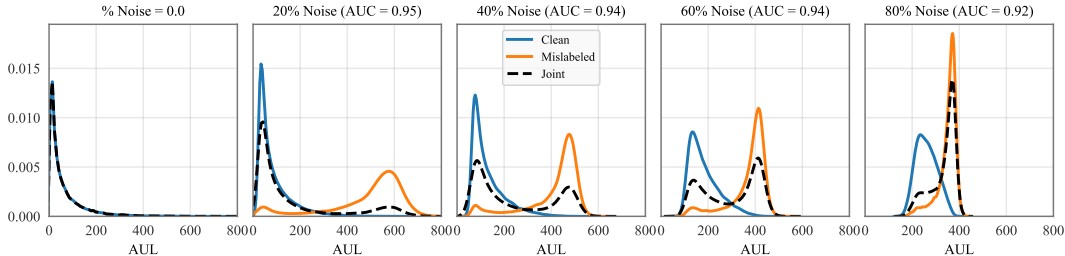

Figure 6: AUL distribution on CIFAR10 (ResNet-32) with varying amounts of mislabeled data. The AUL distribution is bimodal, with the two modes corresponding to clean and mislabeled samples.

distributions using Bayes' rule. Without knowing which samples are clean, estimating these two conditional distributions is an unsupervised problem.

**Approximating the AUL distribution with a Gaussian mixture.** We observe in Figure 6 that the AUL distribution $p(\text{AUL}_{\mathbf{x},\widetilde{y}})$ is distinctly bimodal when label noise is present. The networks in this figure are ResNet-32 models trained on CIFAR10, and the AUL is measured before the learning rate drop (epoch 150). When no noise is present (far left plot), the distribution is unimodal with a long tail of high AUL samples. Intuitively, the mode corresponds to the loss for "easy" or "average" examples while the tail corresponds to "hard" examples. For noisy datasets however, we observe that the separability of AUL creates a distribution with two distinct modes – corresponding to $p(\text{AUL}_{\mathbf{x},\widetilde{y}} \mid y = \widetilde{y})$ and $p(\text{AUL}_{\mathbf{x},\widetilde{y}} \mid y \neq \widetilde{y})$. Though the two modes become closer as the amount of noise increases (*e.g.* with $80\%$ noise the distance between the two modes is a quarter of the distance between $20\%$ noise modes), they are always separable.

We identify these two modes by fitting the AUL distribution with a mixture of two Gaussians. The low-mean Gaussian (blue) approximates the clean-conditional AUL distribution $p(\text{AUL}_{\mathbf{x},\widetilde{y}} \mid y = \widetilde{y})$, while the high-mean Gaussian (orange) approximates the mislabeled distribution $p(\text{AUL}_{\mathbf{x},\widetilde{y}} \mid y \neq \widetilde{y})$. In practice the actual conditional distributions are not necessarily Gaussian, yet we find that this simple model is nevertheless quite good for the applications in Section 5. From this Gaussian mixture, it is easy to estimate the label noise and mislabeled samples. Let $z_{\mathbf{x},\widetilde{y}}$ be the probability that sample $(\mathbf{x}, \widetilde{y})$ belongs to the high-mean (*i.e.* mislabeled) Gaussian distribution. $z_{\mathbf{x},\widetilde{y}}$ is the model's estimate of $p(y \neq \widetilde{y} \mid \text{AUL}_{\mathbf{x},\widetilde{y}})$, and $\mathbb{E}_{\mathbf{x},\widetilde{y}}[z_{\mathbf{x},\widetilde{y}}]$ is the estimate of dataset noise.

### 4.2 Improving Training with Corrupted Labels

A high-capacity model trained on a noisy dataset is in danger of overfitting, as memorizing mislabeled samples may hurt the generalization of correctly labeled samples. Next we describe how to remedy this by using AUL probability estimates to down-weight mislabeled samples during training.

Assume that we have trained a neural network and computed $\text{AUL}_{\mathbf{x}_i,\widetilde{y}_i}$ for every training sample $(\mathbf{x}_i, \widetilde{y}_i)$. Since this network was trained on noisy data it has likely learned features that are specific to the mislabeled examples and will not generalize well. Therefore we propose to train a new model where the samples are reweighed using the AUL probability estimates. More formally, let $z_i = p(\widetilde{y}_i \neq y \mid \text{AUL}_{\mathbf{x}_i,\widetilde{y}_i})$ be the probability that a sample is mislabeled based on the Gaussian mixture fit. We train a network from scratch where the loss for each minibatch $\mathcal{M}$ is computed as

$$\mathcal{L}(\mathcal{M}) = \frac{1}{c} \sum_{i=1}^{|\mathcal{M}|} z_i \mathcal{L}(\mathbf{x}_i, \widetilde{y}_i), \qquad c = \sum_{i=1}^{|\mathcal{M}|} z_i,$$

where $(\mathbf{x}_i, \widetilde{y}_i)$ are the samples in $\mathcal{M}$.

## 5 Results

We measure the efficacy of AUL dynamics at estimating dataset noise, identifying mislabeled samples, and improving training with noisy data. In order to quantitatively measure the performance, we

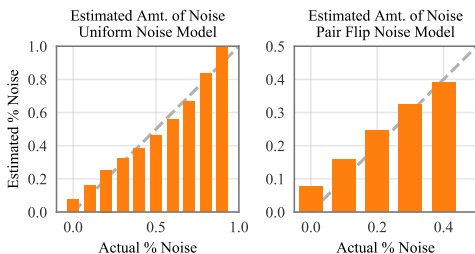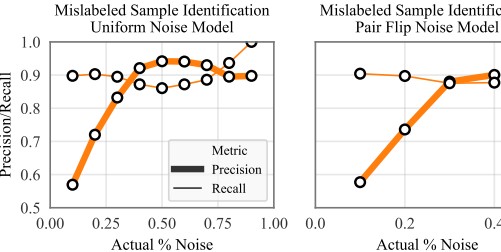

Figure 7: Identifying mislabeled data in noisy variants of CIFAR10 using AUL from trained ResNet-32 models. **Left:** predicted amount of noise in training set (uniform noise model, far left; pair flip noise model, middle left). **Right:** prediction/recall curves for identifying mislabeled data (uniform, middle right; pair flip, far right).

Table 1: Test-error comparison of noisy-data training methods on datasets with corrupted training labels. All methods train a ResNet-32 on the CIFAR10/CIFAR100 datasets with varying amounts/types of label noise. (*Uniform*: mislabeled samples are assigned a random incorrect label. *Pair Flip*: mislabeled samples are assigned a to a class-specific incorrect label.)

| Dataset | CIFAR10 | | | | | CIFAR100 | | | | |
|---|---|---|---|---|---|---|---|---|---|---|
| Noise Model | Uniform | Uniform | Uniform | Uniform | Pair Flip | Uniform | Uniform | Uniform | Uniform | Pair Flip |
| % Mislabeled | 0.20 | 0.40 | 0.60 | 0.80 | 0.40 | 0.20 | 0.40 | 0.60 | 0.80 | 0.40 |
| Standard | $23.2 \pm 0.2$ | $40.3 \pm 0.5$ | $57.1 \pm 0.5$ | $75.3 \pm 0.5$ | $40.3 \pm 0.2$ | $50.3 \pm 0.3$ | $62.0 \pm 0.7$ | $76.0 \pm 0.1$ | $91.1 \pm 0.7$ | $61.8 \pm 0.6$ |
| Random | $16.5 \pm 0.7$ | $22.8 \pm 1.6$ | $35.8 \pm 2.8$ | $55.6 \pm 5.5$ | $21.2 \pm 0.1$ | $43.0 \pm 0.1$ | $51.2 \pm 0.5$ | $63.5 \pm 0.9$ | $83.7 \pm 1.8$ | $51.4 \pm 0.4$ |
| Reed-Hard | $22.4 \pm 0.4$ | $37.4 \pm 0.9$ | $52.0 \pm 0.5$ | $68.8 \pm 0.9$ | $37.2 \pm 0.2$ | $48.6 \pm 0.3$ | $58.9 \pm 0.5$ | $70.3 \pm 0.4$ | $89.8 \pm 1.6$ | $58.8 \pm 0.7$ |
| MentorNet-PD | $13.3 \pm 0.2$ | $18.1 \pm 0.4$ | —— | $66.0 \pm 4.9$ | —— | $35.8 \pm 0.6$ | $42.5 \pm 0.4$ | —— | $\mathbf{75.7 \pm 1.4}$ | —— |
| Co-Teaching | $11.2 \pm 0.2$ | $13.5 \pm 0.2$ | $19.3 \pm 0.2$ | $80.7 \pm 1.2$ | $16.9 \pm 0.6$ | $35.9 \pm 0.2$ | $\mathbf{39.8 \pm 0.4}$ | $52.0 \pm 0.5$ | $89.1 \pm 1.4$ | $51.7 \pm 0.4$ |
| D2L | $12.3 \pm 0.5$ | $15.6 \pm 0.6$ | $27.3 \pm 1.2$ | Diverged | $23.6 \pm 3.0$ | $46.0 \pm 2.0$ | $70.3 \pm 3.6$ | Diverged | Diverged | $83.1 \pm 2.3$ |
| INCV | $10.5 \pm 0.2$ | $13.2 \pm 0.2$ | $18.9 \pm 0.5$ | $46.7 \pm 3.8$ | $16.1 \pm 1.0$ | $41.4 \pm 0.9$ | $44.6 \pm 0.4$ | $56.3 \pm 0.6$ | $\mathbf{76.3 \pm 1.1}$ | $52.0 \pm 0.6$ |
| AUL | $\mathbf{9.2 \pm 0.1}$ | $\mathbf{11.5 \pm 0.2}$ | $\mathbf{16.0 \pm 0.3}$ | $\mathbf{37.8 \pm 2.4}$ | $\mathbf{11.1 \pm 0.3}$ | $\mathbf{34.2 \pm 0.4}$ | $39.9 \pm 0.6$ | $\mathbf{50.5 \pm 1.0}$ | $76.5 \pm 0.9$ | $\mathbf{40.2 \pm 0.2}$ |
| Oracle | $9.0 \pm 0.1$ | $9.7 \pm 0.8$ | $10.8 \pm 1.8$ | $12.6 \pm 3.5$ | $10.8 \pm 0.1$ | $35.5 \pm 0.1$ | $39.0 \pm 0.4$ | $44.8 \pm 1.0$ | $55.1 \pm 0.7$ | $38.6 \pm 0.7$ |

corrupt the labels of the CIFAR10 and CIFAR100 datasets according to two predefined noise models. The **uniform** noise model assumes that discrepancies between true labels and observed labels are random. A given label can be changed to any other class label with equal probability. The **pair flip** noise model assumes that mislabeling mistakes depend on the true label. If an observed label is incorrect, then it will belong to only one other possible class. More formally, if $p$ is the percentage of mislabeled points, we have

$$p_{\text{Uniform}}(\widetilde{y} \mid y) = \begin{cases} (1-p) & \widetilde{y} = y \\ \frac{p}{k-1} & \widetilde{y} \neq y \end{cases}, \qquad p_{\text{Flip}}(\widetilde{y} \mid y) = \begin{cases} (1-p) & \widetilde{y} = y \\ p & \widetilde{y} = (y+1) \\ 0 & \text{o.w.} \end{cases}.$$

These noise models, which have been well-studied in prior work (Han et al., 2018; Hendrycks et al., 2018), represent two extremes: uniform is class-independent and pair flip is fully class-dependent.

We first test the ability of AUL to estimate the amount of mislabeled data in the training set. We construct noisy versions of CIFAR10 and CIFAR100 by applying the two noise models with varying amounts of noise on $45,000$ training examples. The uniform noise model is applied with $10\% - 90\%$ of samples mislabeled, and the pair flip model is applied with $10\% - 40\%$. To compute AUL we train a 32-layer ResNet (He et al., 2016) on each corrupted dataset. The networks are trained for 150 epochs using SGD with a constant learning rate of $0.1$ and a batch size of $64$. This corresponds to the standard training hyperparameters for ResNets before the first learning rate drop (which typically occurs at epoch 151). We identify potentially mislabeled samples by fitting the AUL distribution with a mixture of 2 Gaussians, as described in Section 4.1. Results are averaged over 4 trials.

**Noise estimates.** Figure 7 displays the estimated noise for the corrupted CIFAR10 datasets (uniform noise – left, pair flip – middle left). We observe that AUL tends to overestimate high in the low- and high-noise settings (e.g. below $0.2$ and above $0.8$ in the uniform noise setting). Nevertheless, the AUL-based estimates are able to closely recover the true amount of noise in most settings. Analogous plots for the corrupted CIFAR100 datasets can be found in Appendix A.

**Identifying mislabeled samples.** After fitting the Gaussian mixture to the AUL distribution, we classify examples as mislabeled if $p(\widetilde{y} \neq y \mid \text{AUL}_{\mathbf{x}, \widetilde{y}}) > 0.5$. Figure 7 displays the precision

and recall of mislabeled sample identification on the corrupted CIFAR10 datasets (uniform noise – middle right, pair flip – right). (See Appendix A for CIFAR100 plots.) The predictions achieve a recall of roughly $90\%$ on most noise models, suggesting that the AUL distribution model identifies most mislabeled points. The AUL estimates have the lowest precision in the small noise regime – around $60\%$ to $70\%$. However, this corresponds to relatively few misclassifications since the number of predicted mislabeled samples is small in this setting. In the other noise settings the model achieves precision around $90\%$. It is worth emphasizing that the AUL model achieves this high performance *without any supervision or prior knowledge about the noise model.*

**Robust training with corrupted labels.**    Next we evaluate the performance of models trained on corrupted datasets with the reweighting scheme proposed in Section 4.2. In all experiments, we train 32-layer ResNets on the CIFAR10 and CIFAR100 variants described above. We assume that we do not have access to a clean validation set, and so all models are trained without early stopping.

To set up a sensible lower bound for performance, we train a network that follows a **Standard** training procedure (i.e. no method for combating mislabeled samples). The **Oracle** baseline – which is only trained on correctly-labeled portion of data – represents a sensible upper bound. We also compare to a **Random** (Ren et al., 2018) baseline, where each sample is assigned a random weight. The weights are drawn from a rectified normal distribution Ren et al. (2018) and are re-drawn at every epoch. **Reed-Hard** (Reed et al., 2014) and **D2L** (Ma et al., 2018) interpolate the one-hot target label with the predicted label. **MentorNet-PD** (Jiang et al., 2017) learns a weighting scheme for training examples using a LSTM. [1] **Co-teaching** (Han et al., 2018) simultaneously train two networks that inform each other about which training examples to keep. **INCV** (Chen et al., 2019) is an extension of co-teaching that first filters training data through iterative cross-validation.

We train all networks on $45,000$ training examples. We compute the AUL of these samples by training a network for 150 epochs with a learning rate of $0.1$. AULs are converted to sample weights using the Gaussian mixture described in Section 4.2. The standard, oracle, and reweighted networks are trained using the procedure of He et al. (2016) – 300 epochs of SGD with an initial learning rate of $0.1$ that is dropped by a factor of 10 at epochs 150 and 225. All other methods are trained using hyperparameters that are defined in their publicly available implementations.

Table 1 displays the test set performance for the methods on the corrupted CIFAR10 and CIFAR100 datasets. We observe several trends. First, there is a large performance gap between the Oracle and Standard baselines, suggesting that the standard networks overfit to the mislabeled examples. On CIFAR10 the gap is $14\%$ with $20\%$ uniform noise, and $62\%$ with $80\%$ noise. Most of the baseline methods reduce this performance gap. Our proposed reweighting scheme (AUL) achieves the best error on all dataset variants, especially on the high uniform noise and pair flip datasets. Most notably, AUL is able to recover the oracle performance on low-noise CIFAR10 and surpass oracle performance on low-noise CIFAR100. We hypothesize that the AUL reweighting not only identifies mislabeled examples but also reduces the impact of difficult or ambiguous examples.

## 6    DISCUSSION AND CONCLUSION

In this paper we introduce the AUL statistic, which reveals differences in training loss dynamics to reliably identify noisy labels. We observe that this statistic is far more stable across architectures than for example the validation loss. This finding is compatible with the commonly made observation that even networks with different initializations vary noticeably in their predictions — a welcome side-effect for model ensembling (Caruana et al., 2004), but a nuisance when detecting outliers. While the AUL score should be highly effective to increase classifier robustness and analyze or improve the label quality of data sets, we believe that the its ability to detect samples with high memory need may open applications far beyond the scope of this paper. As future work, we are planning to investigate the use of the AUL score to estimate sample "hardness" for the use in curriculum learning (Bengio et al., 2009) and to detect mis-predictions on unsupervised data in the context of semi-supervised learning (Chapelle et al., 2009).

---

[1]Due to incomplete release, we cannot train MentorNet (Jiang et al., 2017) in few noise settings.

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

# A   ADDITIONAL RESULTS

## A.1   AVERAGE LOSS AND AUL TRAJECTORIES

To further visualize the difference between the loss and AUL metrics, we plot the distribution of loss and AUL as a function of training epoch in Figure 8. The losses and AULs are computed using a ResNet-20 model trained on CIFAR10 with $40\%$ uniform noise. We see that the loss distributions for clean and mislabeled data points become less separable as training increases. Conversely, AUL remains separable throughout the course of training.

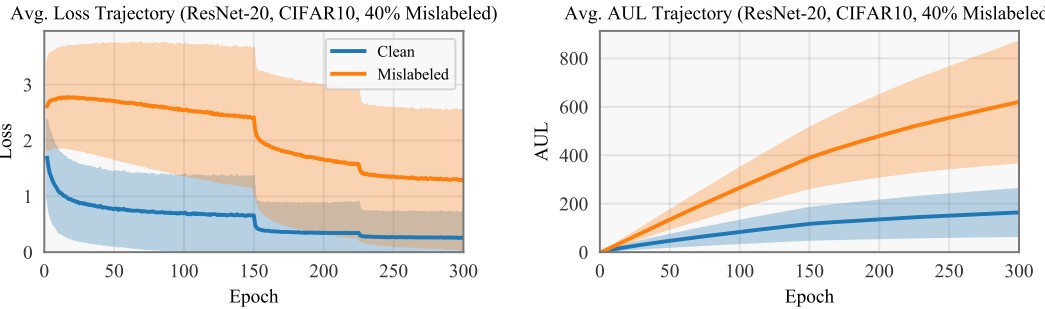

Figure 8: Distribution of loss (left) and AUL (right) over the course of network training. Shaded region represents 1 standard deviation.

## A.2   NOISE ESTIMATES AND MISLABELED SAMPLE IDENTIFICATION FOR CIFAR100

In this section we replicated the experiments from Section 5 on the CIFAR100 dataset. Figure 9 (left, middle left) displays the noise estimation for the CIFAR100 dataset on the uniform and pair flip models, respectively. We do note that the estimates for the pair flip noise models tend to be larger than the actual amount of noise. Nevertheless, we find that – similarly to CIFAR10 – the model is able to accurately recover the amount of noise in the dataset. Figure 9 (middle right, right) displays the identification precision/recall for the CIFAR100 dataset. We again find that the recall is in the upper $90\%$. The precision tends to be lower with less noise, but rises above $80\%$ when there is more than $30\%$ noise.

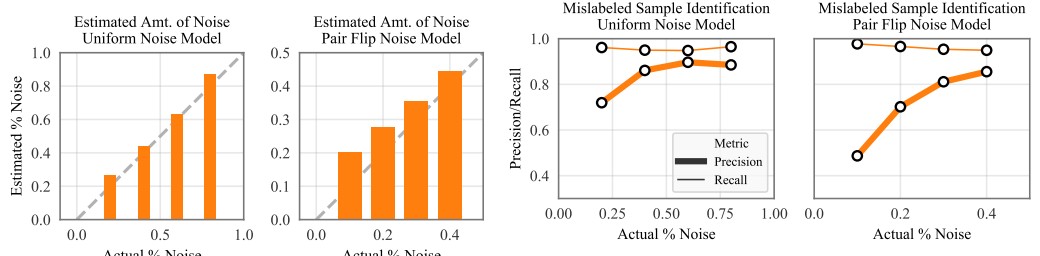

Figure 9:   Identifying mislabeled data in noisy variants of CIFAR100 using AUL from trained ResNet-32 models. **Left:** predicted amount of noise in training set (uniform noise model, far left; pair flip noise model, middle left). **Right:** prediction/recall curves for identifying mislabeled data (uniform, middle right; pair flip, far right).

## A.3 ADDITIONAL HIGH-AUL IMAGES

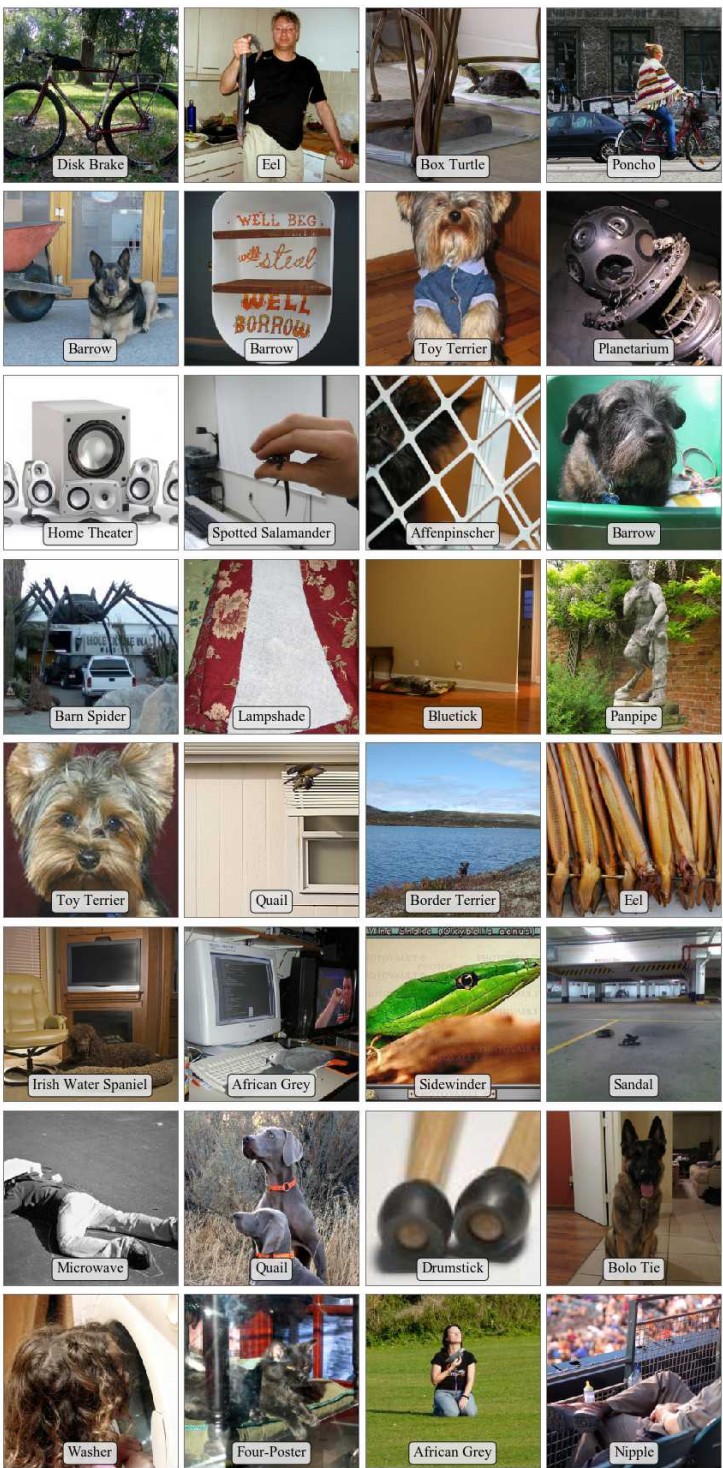

Figure 10: Additional images from the first 100 classes of ImageNet with large AUL when trained on a DenseNet-121 model.

