# OpenReview forum: "Detecting Noisy Training Data with Loss Curves"
_ICLR.cc/2020/Conference — Reject_

### Official Review · AnonReviewer1 · 2019-10-19
**Official Blind Review #1**

**Rating:** 3

**Review:**

Pros:
1. This work suggests a surprisingly elegant approach to identify data with corrupted labels. The proposed AUL (area under the loss curve) makes intuitive sense, and exploiting the fact that noisy examples are hard to learn at the early stage of training, leading to higher AULs than clean examples.
2. The paper is well-written, with plenty of visualization to aid the understanding.
3. Experimental results suggest advantage of the proposal. Evaluation seem to suggest the using AULs is more robust against large corruptions compared with SOTA.

Cons:
1. The biggest concern is the generalizability of AULs to datasets that are more challenging than CIFAR10 and CIFAR100. This has been the concern not only for this work, but also for this general research area. For instance, on Tiny-ImageNet dataset (e.g., used in SELFIE: Refurbishing Unclean Samples for Robust Deep Learning (ICML19), which could btw be used in the benchmark), a clean dataset only gives ~55% test error, compared with about ~10% on CIFAR10 and ~30% on CIFAR100. The robustness of AULs needs to be tested on those harder problems as well.
2. AULs are based on the assumption of a clear separation of two types of examples -- clean and noisy. It ignores the difficulty of training examples. This raises the question of where to place hard-clean examples between those two modes.
3. It usually helps to include a real-world noisy datasets where its noisy corruption is unknown. One would be curious to know the noisy level estimate.

Questions:
1. It seems from the experiments that MoGs are fit per training example in order to compute weights for the cost?
2. The computational cost of AUL-based training is not mentioned. Is it 2x, 3x, considering there are distinct stages of training before and after AUL estimation?

**Experience Assessment:**

I have read many papers in this area.

**Review Assessment: Checking Correctness Of Derivations And Theory:**

I carefully checked the derivations and theory.

**Review Assessment: Checking Correctness Of Experiments:**

I carefully checked the experiments.

**Review Assessment: Thoroughness In Paper Reading:**

I read the paper thoroughly.

---

> ### Author Response · Authors · 2019-11-11
> **Response from authors**
>
> W.R.T. additional benchmarks and datasets: We have since performed additional experiments on TinyImagenet. A ResNet-32 trained on 95000 training samples (5000 withheld for validation) receives 49.4% test error on the clean dataset. With 40% label noise this number drops to 65.5%. AUL reweighting achieves 55.7% error. We will update the paper with these results. Additionally, see the response to Reviewer #2 regarding real-world datasets.
>
> W.R.T. “hard” clean examples: please see response to Reviewer #2.
>
> W.R.T. the MoG: we use the initial network to compute the AUL for all 45,000 training points. Then we fit the MoG to these 45,000 AULs.
>
> W.R.T the computation cost: thank you for noting this omission. It is roughly 1.5x as much as normal training. The initial AUL estimate requires training a network for 0.5x the usual training budget (right before the learning rate is dropped for the first time) Fitting the MoG is negligible compared to training the network.

---

### Official Review · AnonReviewer2 · 2019-10-22
**Official Blind Review #2**

**Rating:** 3

**Review:**

In this paper, the authors proposed methodologies to identify mislabeled training examples. Specifically, the authors proposed to examine the training loss of each example to identify examples with high training losses. To mitigate the randomness in the training loss, the authors proposed an AUL (area under the loss) metric, which is less noisy that directly examining the loss. The authors proposed to use Gaussian mixture to approximate the AUL distribution, and then estimate the probability of mislabeling. The authors proposed to downweight training examples with high likelihood of being mislabeled. The proposal is examined in numerical studies.

While I agree the method has a good potential to be significant, in its current form, I have several concerns that prevent me from recommending acceptance.
1. How can we be sure that examples with large training loss are mislabeled examples, rather than examples that are inherently difficult to classify? For example, if the dataset consists of 99% bright images and 1% dark images, then it could be possible that dark image training examples have a higher loss. This question becomes more relevant when the model does not have sufficient complexity, or does not have the correct form of complexity.
2. If we cannot distinguish mislabeled examples with examples that are hard to classify, then it seems to me that a simple downweighting based on AUL could potential lead to the opposite outcome. Intuitively, we should upweight those difficult to classify examples to improve generalizability.
3. It is somewhat surprising to me that the AUL distribution is a nicely looking bimodal shape. Could it be due to the fact that the wrong labels are randomly generated with equal probabilities from other classes? What it we have a different error distribution, e.g., the label has p probability of being correct, and (1-p) probability of being k, where k is a constant?
4. Given the above concerns, I feel the numerical studies are insufficient. The authors may consider 1) including more noise distributions, 2) presenting the AUL distributions to check whether they are truly bimodal, and 3) trying more datasets and more model architectures.

**Experience Assessment:**

I have read many papers in this area.

**Review Assessment: Checking Correctness Of Derivations And Theory:**

I carefully checked the derivations and theory.

**Review Assessment: Checking Correctness Of Experiments:**

I assessed the sensibility of the experiments.

**Review Assessment: Thoroughness In Paper Reading:**

I read the paper at least twice and used my best judgement in assessing the paper.

---

> ### Author Response · Authors · 2019-11-11
> **Response from authors**
>
> Thank you for your review. We hope that we are able to clarify some misunderstanding and address your comments.
>
> W.R.T. “hard” clean examples: From our observations, it appears that “hard” examples have higher AUL than most clean examples, but lower AUL than mislabeled examples. (Intuitively, this makes sense, because truly mislabeled examples will be the hardest for the network to generalize to.) Therefore, hard samples will be downweighted some, but not as much as noisy examples. We would note that curriculum learning (e.g. Bengio et al, ICML 2019; Hacohen and Weinshall, ICML 2019) similarly down-weights (rather than up-weights) difficult examples until the end of training. To some extent AUL-based downweighting will have a similar effect: the network will not reduce the loss of difficult examples until the high-weight easy examples have sufficiently low loss.
>
> W.R.T. other noise models: Please refer to our “pair flip” experiments in Figure 7 and Table 1. These experiments are exactly the noise model you propose (p probability of being correct, and (1-p) probability of being k). We note that our proposed method is adept at estimating and correcting for label error in this setting as well.
>
> W.R.T. presenting AUL distributions: we observed the bimodal behavior for both CIFAR10 and CIFAR100 on all experiments that we performed - regardless of noise model, noise percentage, or network architecture. We will update the paper with additional plots that demonstrate this bimodality.
>
> W.R.T. real-world datasets: To demonstrate our method’s applicability on real-world (non-image) datasets, we trained a model on a Machine Translation dataset (IWSLT14 German-to-English dataset). AUL determines that roughly 25 training samples have an incorrect target (the English sentence in the translation pair). Typically, these targets are not English and therefore are mislabeled. For example:
>
> SOURCE: vielen dank .
> TARGET: merci beaucoup .
>
> SOURCE: google glass : aroi mann 9 : mmm , aroi .
> TARGET: google glass : อร ่ อยman 9 : mmm , อร ่ อย .

---

### Official Review · AnonReviewer3 · 2019-10-23
**Official Blind Review #3**

**Rating:** 3

**Review:**

This paper proposes a new method of detecting label noise through calculating the Area Under the Loss statistic, which is based on the learning trajectory.

I believe this paper is NOT the first one to point out the different training behavior of clean and noisy samples in the label noise problem. While the author mentioned that Shen & Sanghavi used the training losses for selecting data, Shen & Sanghavi also observed the training loss for good and bad samples are different across time. Therefore, it would be good to give correct credit to previous work, and not over-emphasizing the contribution.

Please proofread the paper, and correct the typos. For example, on page 3, there are 2 typos in the last three lines in the paragraph “rate of loss reduction indicates memorization need”.

Can you explain why you normalize the loss of each batch by c ? If z_i s are all small within a batch, this loss function does not reweight the bad samples in the correct way.

Can you explain why you use Pearson’s correlation instead of Spearman’s correlation to evaluate the consistency of the ranking? This sounds surprising to me.

Also, the cited paper “Understanding generalization of deep neural networks trained with noisy labels. “ by Hu et al. is not in NeurIPS 2019. Please correct this citation error if possible.

**Experience Assessment:**

I have published one or two papers in this area.

**Review Assessment: Checking Correctness Of Derivations And Theory:**

I assessed the sensibility of the derivations and theory.

**Review Assessment: Checking Correctness Of Experiments:**

I assessed the sensibility of the experiments.

**Review Assessment: Thoroughness In Paper Reading:**

I read the paper at least twice and used my best judgement in assessing the paper.

---

> ### Author Response · Authors · 2019-11-11
> **Response from authors**
>
> Thank you for your review. We answer some of your questions below. (Additionally, thank you for catching typos and the NeurIPS 2019 citation error. We will correct these.)
>
> W.R.T. existing work: We will ensure that appropriate credit is given to Shen & Sanghavi in the revised version. We would note that our contribution goes beyond identifying the differences in training loss (also noted by e.g. Zhang et al., ICLR 2017), but introducing a stable metric (AUL) that is able to transform this observation into a robust method.
>
> W.R.T. normalizing by c: This is a common practice for several re-weighting methods (e.g. Ren et al., ICML 2018). We found that it improves the stability of training as it ensures that the effective learning rate is the same for all datasets regardless of label noise. We would note that in practice it would be incredibly rare for a minibatch to contain only bad examples. Even with 80% label noise, the probability of an all bad minibatch of size 256 is roughly 10^-25.
>
> W.R.T. Pearson’s vs Spearman’s: Spearman’s produces similar results (e.g. ~98% correlation for AUL across networks). We chose to measure the Pearson’s correlation because—-as noted in Figure 4—-the AUL scores from each network fall within very similar ranges. Therefore we expect correlation among the values and not just their rankings. We will revise the language in the Figure 5 caption which may have caused confusion.

---

### Decision · Program_Chairs · 2019-12-19

**Decision:**

Reject

**Comment:**

The paper proposes a new, stable metric, called Area Under Loss curve (AUL) to recognize mislabeled samples in a dataset due to the different behavior of their loss function over time. The paper build on earlier observations (e.g. by Shen & Sanghavi) to propose this new metric as a concrete solution to the mislabeling problem.

Although the reviewers remarked that this is an interesting approach for a relevant problems, they expressed several concerns regarding this paper. Two of them are whether the hardness of a sample would also result in high AUL scores, and another whether the results hold up under realistic mislabelings, rather than artificial label swapping / replacing. The authors did anecdotally suggest that neither of these effects has a major impact on the results. Still, I think a precise analysis of these effects would be critically important to have in the paper. Especially since there might be a complex interaction between the 'hardness' of samples and mislabelings (an MNIST 1 that looks like a 7 might be sooner mislabeled than a 1 that doesn't look like a 7). The authors show some examples of 'real' mislabeled sentences recognized by the model but it is still unclear whether downweighting these helped final test set performance in this case.

Because of these issues, I cannot recommend acceptance of the paper in its current state. However, based on the identified relevance of the problem tackled and the identified potential for significant impact I do think this could be a great paper in a next iteration.